# Application of Nanopore Sequencing in the Detection of Foodborne Microorganisms

**DOI:** 10.3390/nano12091534

**Published:** 2022-05-02

**Authors:** You Zhou, Meishen Ren, Pengfei Zhang, Dike Jiang, Xueping Yao, Yan Luo, Zexiao Yang, Yin Wang

**Affiliations:** Key Laboratory of Animal Diseases and Human Health of Sichuan Province, College of Veterinary Medicine, Sichuan Agricultural University, Chengdu 611130, China; 2021203043@stu.sicau.edu.cn (Y.Z.); 14773@sicau.edu.cn (M.R.); 2018103006@stu.sicau.edu.cn (P.Z.); 2020103005@stu.sicau.edu.cn (D.J.); 13577@sicau.edu.cn (X.Y.); 41187@sicau.edu.cn (Y.L.); 13643@sicau.edu.cn (Z.Y.)

**Keywords:** foodborne diseases, nanopore sequencing technology, WGS, metagenomics, real-time monitoring, poultry health, COVID-19

## Abstract

Foodborne pathogens have become the subject of intense interest because of their high incidence and mortality worldwide. In the past few decades, people have developed many methods to solve this challenge. At present, methods such as traditional microbial culture methods, nucleic acid or protein-based pathogen detection methods, and whole-genome analysis are widely used in the detection of pathogenic microorganisms in food. However, these methods are limited by time-consuming, cumbersome operations or high costs. The development of nanopore sequencing technology offers the possibility to address these shortcomings. Nanopore sequencing, a third-generation technology, has the advantages of simple operation, high sensitivity, real-time sequencing, and low turnaround time. It can be widely used in the rapid detection and serotyping of foodborne pathogens. This review article discusses foodborne diseases, the principle of nanopore sequencing technology, the application of nanopore sequencing technology in foodborne pathogens detection, as well as its development prospects.

## 1. Introduction

The incidence of foodborne diseases is gradually increasing, which has aroused people’s attention to food safety. One of the main causes of foodborne illness is eating food contaminated with bacteria, viruses, or toxins such as Campylobacter, Salmonella, Listeria, Norovirus, etc. [1]. According to the World Health Organization WHO [2], around 600 million foodborne illnesses and 420,000 deaths occur globally each year associated with various pathogens and toxins in food products. The occurrence of foodborne disease is regional [3], with the highest incidence of foodborne illness in Africa. This is due to a number of factors, including the lack of a safe food supply chain [4]. The lack of effective detection of foodborne pathogens is also a major contributor to foodborne illness [5]. However, foodborne diseases pose a huge threat to public health, even in countries with well-developed food supply chains and infectious disease surveillance systems [6]. In the US, FoodNet reported 25,866 cases of infection, 6164 hospitalizations, and 122 deaths in 2019 [7]. In addition, some patients who show milder symptoms never arrive at a proper diagnosis. This means that foodborne diseases are often severely underestimated.

In order to handle this global challenge, rapid detection of foodborne pathogens is needed. In the last decades, some effective methods have been developed for the rapid detection of foodborne pathogens in food products [8]. Currently, in addition to conventional bacterial culturing, other methods such as nucleic acid and antibody methods are also widely used in the detection of foodborne pathogens [9]. However, all of these methods have limitations, as shown in Table 1. Recently developed nanopore platforms have opened up possibilities for the detection of foodborne pathogens. Nanopore platforms are an innovative technology in the fields of nanobiodevices and structural biology [10] that can differentiate various types of cancer cells and bacteria by shape, thus enabling their direct identification [11]. Through machine learning, the nanopore platform could be an accurate way to identify individual bacteria in real-time [12].

DNA can provide more interesting information from the perspective of preventing and controlling foodborne diseases. The detailed genetic information not only facilitates accurate detection and typing of pathogens but can also support their source attribution through phylogenetic analysis.

Advances in sequencing technologies have facilitated whole-genome sequencing (WGS) and metagenomic analysis [13]. Sequencing has emerged as a new strategy for food defense [14]. The second-generation sequencing represented by Illumina provides unique advantages for the detection and identification of foodborne pathogens, including serotyping, outbreak source tracking, phylogenetic localization, and so on. There are several remaining limitations mentioned below, although second-generation sequencing has been broadly used to detect and identify foodborne pathogens. Second-generation sequencing has disadvantages such as low throughput, the inability of real-time analysis, and time-consuming library preparation methods. In addition, second-generation sequencing instruments are expensive and non-portable, which is economically prohibitive in the food industry [15]. In recent years, third-generation technology has improved greatly. The nanopore sequencing technology developed by Oxford Nanopore Technologies has its own unique advantages, such as long reading, real-time sequencing, and portability, and it has become a promising method for various fields [16,17]. In particular, nanopore sequencing has been exploited for its rapid detection and single-molecule sequencing capability in the design strategy of foodborne pathogens detection [18].

This review first introduces the principles of nanopore sequencing and subsequently describes its application in several areas of foodborne disease research. Finally, current challenges and possible future improvements in nanopore sequencing are discussed.

## 2. Nanopore Sequencing Platform

The detection of pathogenic microorganisms that may exist in food processing links and food is an effective means of preventing foodborne diseases [33]. On the other hand, emerging foodborne illness findings can be linked to contaminated food. This connection will facilitate the removal of contaminated food from the food chain and prevent further contamination by foodborne pathogens [34]. Compared to traditional methods, sequencing technology offers new advantages. In addition to identifying foodborne pathogens, whole-genome sequencing can also obtain information about virulence, drug resistance, serotype, and other information [35]. Using metagenomic sequencing, unknown or difficult-to-cultivate pathogens can be identified without culturing the pathogens, as with traditional culture methods [36]. However, sequencing strategies have not been applied to the detection of foodborne pathogens on a large scale. One of the reasons is that the first- and second-generation sequencing instruments, represented by Sanger sequencing and Illumina sequencing, respectively, are often expensive, and the cost of sequencing is high, making it difficult to apply on a large scale in some economically underdeveloped countries and regions. Moreover, the operation steps required for first- and second-generation sequencing are cumbersome, time-consuming, and low-throughput, which cannot be adapted to the rapid detection of outbreaks of foodborne diseases [37]. Emerging nanopore sequencing technologies overcome these limitations and provide new ideas for the rapid detection of foodborne pathogens.

### The Principle of Nanopore Sequencing

The idea of using nanopores for sequencing was first proposed in 1980 [38]. After a long period of experimentation, Oxford Nanopore Technologies (ONT) released its first commercial sequencing device, the MinION, in 2013. The basic principle of nanopore sequencers is to determine the bases by measuring the instantaneous potential changes caused by molecules passing through nanopores [39]. By connecting the nanopore sequencer to a computer, a fixed electric field is formed at both ends of the instrument, which makes the ions in the buffer move to form a stable current. DNA and other molecules will move towards the nanopore under the effect of an electric field after they are dropped into the sequencer. Molecules passing through the nanopore cause a brief blockage, perturbing the steady-state current flow. Due to the difference in the structure and size of the molecules, the induced transient current will also change accordingly. For example, when a DNA molecule approaches a nanopore, the motor protein added at the time of library preparation unwinds two strands of DNA into one. The single-strand then rushes through the nanopore under an electric field. Due to the differences in molecular structure and molecular mass of the four bases of ATCG, the current induced by blocking shows characteristic changes. The detectors built into the sequencer can detect these characteristic fluctuations and transmit them to the computer. Then, the computer analyzes the characteristic signals of various bases and converts them into the base sequence (Figure 1).

## 3. Application of Nanopore Sequencing in Foodborne Disease

Nanopore sequencing identifies base sequences using electrical signals, which means it has the ability to directly detect individual molecules such as DNA, RNA, and peptides [40,41]. Compared to traditional second-generation sequencing, the principle of nanopore sequencing determines its unique advantages. Nanopore sequencing technology has the advantages of high throughput, low cost, and short-time library preparation. The potential of nanopore sequencing for the detection and identification of foodborne pathogens has been shown in many different studies (Table 2). Currently, whole-genome sequencing and metagenomics analysis are the two most used methods for nanopore sequencing.

### 3.1. Nanopore Technology-Based WGS for Food Safety

The frequency of global food safety issues has led to higher requirements for the detection of current foodborne diseases. With the development of sequencing technology, WGS has become a powerful tool for the detection of foodborne pathogens [56]. Today, several public agencies, such as Food and Drug Administration (FDA), Centers for Disease Control and Prevention (CDC), Public Health England, European Center for Disease Prevention and Control (ECDC), are widely using WGS to investigate the occurrence of foodborne diseases [57]. WGS has important application value in the food safety industry because it not only can accurately identify and subtype foodborne pathogens but also use genetic information to pinpoint the identity and origin of microorganisms that contaminate food and cause disease outbreaks [7]. WGS relies on the selection and culture of a single isolate. The genetic information obtained by sequencing can not only help us to identify the pathogenic pathogen accurately but also can promote the understanding of the pathogen [58].

#### 3.1.1. Rapid Identification and Typing of Foodborne Pathogens

Nanopore sequencing has powerful advantages for rapid WGS analysis of foodborne pathogens. One of the advantages of nanopore sequencing technology is the simple and fast library preparation process [59]. The library preparation process before sequencing directly affects the final sequencing result. The library construction of second-generation sequencing is relatively complex, involving steps such as end modification, adding adapters, purification, and PCR. This greatly increases the preparation time before sequencing. Unlike next-generation sequencing, library preparation for nanopore sequencing is simple and fast. Library construction for nanopore sequencing requires only the addition of motor proteins and adapters. In addition, users can select appropriate library construction kits according to nucleic acid types, PCR, and other conditions, which greatly increases the simplicity and flexibility of library preparation methods. If rapid sequencing is desired, nanopore sequencing enables rapid library preparation without relying on non-portable instruments. This means that nanopore sequencing can complete the rapid library preparation at any location, which is very beneficial for ad hoc detection in non-laboratory settings (Figure 2).

Clear sequence data generated by WGS provides strong evidence for the identification and typing of foodborne pathogens. In the past few decades, short-read sequencing technology has been widely used in the detection of foodborne pathogens. However, when testing for foodborne pathogens, especially bacteria, due to the limitation of sequencing read length, short-read sequencing is not satisfactory for target gene detection, mapping complex repetitive regions, and genome assembly. For example, short-read sequencing may miss some randomly distributed genomic segments, leading to false detection of bacterial virulence factors and resistance genes. The advent of nanopore sequencing has changed that. Nanopore sequencing does not require the chain reaction of DNA polymerase. In theory, the length of the DNA molecule determines the length of the reads. Long reads provide a more definitive way to align and match DNA or RNA sequences, providing high-quality, more complete, and contiguous genome assemblies [60]. It not only reduces the time required for splicing but also provides higher quality data for subsequent genomic analysis. There is evidence that the long reads provided by nanopore sequencing can be used to identify strains within 30 min and to complete typing in the subsequent generation of more detailed sequence data [61]. Sarah Azinheiro et al. [62] verified that nanopore sequencing could be used as a highly sensitive detection method to detect foodborne pathogens. Judgment results obtained by nanopore sequencing have good consistency with the results of qPCR and culture. However, compared to traditional sequencing methods, nanopore sequencing has a higher error rate [63]. This limits the development of this technology in the food industry. Especially in metagenomic analysis, the presence of large amounts of genetic data on host genomes and non-target microorganisms in food matrices may further lead to a decline in data quality and accuracy [64]. However, this does not prevent us from considering nanopore sequencing as a promising method for detecting foodborne pathogens. Several studies have demonstrated the applicability of nanopore sequencing for the analysis of foodborne pathogens.

#### 3.1.2. Bacterial Resistance Detection

The emergence of antimicrobial resistance (AMR) among foodborne pathogens is recognized as a major public health concern, as AMRs may spread along the food chain, leading to widespread acquired resistance to some antimicrobials [65]. Compared to methods such as antimicrobial sensitivity testing and PCR, WGS has now become an invaluable tool for detecting AMR, as shown in Table 3 [66]. The accurate and complete genetic information provided by WGS allows us to search for potential AMR genes. The long reads and high-quality genome sequences provided by nanopore sequencing facilitate the real-time identification of existing or putative novel AMR genes. Jamie K. Lemon et al. [67] using MinION allowed successful identification of antimicrobial resistance genes in the draft assembly corresponding to all classes of observed plasmid-based phenotypic resistance. Moreover, MinION could obtain AMR genes from subcultured isolates within 6 h. This is very promising for monitoring novel and highly divergent AMR genes present throughout the food processing chain.

#### 3.1.3. Detection of RNA and RNA Modifications

The ability to directly detect RNA is a powerful advantage of nanopore sequencing technology [75]. At present, first-generation sequencing and second-generation sequencing require reverse transcription of RNA to cDNA during the library preparation stage. This can lead to the introduction of errors or biases in the sequencing data. Nanopore sequencing technology uses electrical signals to directly sequence RNA, which can effectively avoid the shortcomings of RT-PCR. Moreover, nanopore sequencing allows the detection of RNA modifications and poly-A tail length [76]. This has important biological implications. Nanopore direct RNA sequencing is revolutionary for the rapid detection of foodborne pathogens. First, when detecting foodborne RNA viruses such as norovirus, a suitable RNA library preparation kit for rapid sequencing of the pathogen can be directly selected, reducing the time lost by reverse transcription PCR [77]. Emily Rames and Joanne Macdonald [78] demonstrated that nanopore sequencing provides sequencing results in hours, with potential for gene analysis of enteroviruses in clinical and environmental sources. The short turnaround time is a strong attraction when faced with an outbreak of foodborne illness caused by an RNA viroid infection. This helps us quickly identify foodborne pathogens that cause illness and trace them back to the contaminated food, remove them from the food chain and prevent further infection. Second, some studies have proposed that the microorganisms contained in the food matrix and the stable DNA produced by the death of foreign microorganisms in food processing, transportation, and storage may complicate the genomic background in foods. When using DNA-based assays, the background genome can interfere with the assay, resulting in false-positive results. Therefore, Hellyer et al. [79] proposed that using RNA instead of DNA as a biomarker of bacterial survival may be a better option because RNA has a shorter half-life than DNA in an environment of cell inactivation. Nanopore sequencing has the ability to obtain RNA transcripts. Compared to traditional methods, nanopore transcriptome sequencing offers advantages in read length and handling complex microbiome and nonbacterial transcriptome backgrounds. Yi Fang et al. [80] established a new strategy for RNA library construction and successfully sequenced transcripts of the bacterial pathogen L. monocytogenes using nanopore sequencing. It demonstrates the promise of nanopore technology for real-time multiplex identification of live bacteria in food and is expected to be a detection method using RNA as a marker.

RNA modification has always been a focus of attention. RNA modification involves transcription, transport, translation, and other processes, and plays an important role in cellular physiological activities [81]. There are more than 100 different types of post-synthetic modifications in RNA, and all 4 bases and ribose sugars of RNA can be targets for modification. For instance, N6-methyladenosine (m6A) [82] and 5-methylcytosine (5-mC) [83] are the most common internal modifications in mRNA and play important roles in the metabolism and regulation of various RNAs. Another advantage of nanopore sequencing over previous sequencing methods is the detection of nucleotide modifications. In principle, nanopore sequencing is able to detect any RNA modification of interest at single-nucleotide resolution and in individual full-length native RNAs. Since the presence of modified bases results in an altered ionic current signal from the unmodified base as it passes through the pore, nanopore sequencing is able to detect the modifications without any additional sample preparation. To date, nanopore sequencing has been proven capable of detecting a wide range of RNA modification types, including m6A, N7-methylguanosine (m7G), 5-mC, 5-hydroxymethylcytosine (hm5C), pseudouridine (Y), and 2′-O-methylations (Nm), among others [84].

Nanopore sequencing technology has the potential to facilitate the in-depth study of many biological processes and help uncover the complexities of mRNA processing and modification in various pathogens.

### 3.2. Metagenomics Based on Nanopore Sequencing

Metagenomics refers to the genomes of the total microbiota found in nature [36]. In recent years, the progress of sequencing technology has promoted the development of metagenomics in the field of food safety. Compared with WGS, the advantage of the metagenomic method is that it does not require the pre-cultivation of microorganisms in food and the environment. This means that metagenomic sequencing provides a culture-independent alternative to the direct detection of pathogens in food in the presence of unculturable viruses and parasites. Metagenomics analysis provides the ability to detect and type foodborne pathogens in a single workflow. At the same time, additional data can be generated for in-depth analysis [85]. The new technological advantages brought by nanopore sequencing can enable better application of metagenomics in the field of food safety (Figure 3).

Metagenomics targets all microbial genomes in the environment. While this allows skipping the culture step and directly sequencing the pathogen of interest, it also means that it is necessary to screen for target sequences from a complex genomic background [86]. In fact, this is very difficult. This is especially true for the detection of pathogens in food, as the background genome may contain both microbiota contained in the food matrix and foreign microbiota introduced during food production, food processing, storage, and transport. All stable DNA molecules in the environment are expanded and sequenced. This may overwrite the true sequencing results of the target pathogen. In addition, traditional metagenomic analysis based on second-generation sequencing relies on sequencing depth to obtain sufficient data to identify low-abundance microbial species from the environment. This causes us to waste a lot of time and effort when dealing with the huge amount of data obtained by sequencing. Therefore, when detecting low-abundance foodborne pathogens in food, traditional sequencing methods often face huge challenges in the face of complex genomic backgrounds. Metagenomics analysis based on nanopore sequencing technology has shown excellent performance in the detection of foodborne pathogens [87]. Moreover, some studies also showed that the host DNA in the sample has great significance to the sensitivity of the nanopore metagenomic sequencing [88,89]. Compared to the group without removing host DNA, the nanopore sequencing of the removed group required less time to determine the pathogens present in the sample, and the resulting sequence data was significantly increased. This means that removing the host DNA to reduce the background genome is still necessary [90]. Sensitivity for direct metagenomic analysis using nanopore sequencing can be low, and the error rate of sequencing results increases [91]. Therefore, the target sequence enrichment of specific pathogens in the nanopore sequencing is critical. Particularly, microbial sequences can be drowned out by a large number of non-target sequences when the target pathogen in the metagenomic sample is too low [92]. Therefore, targeted sequencing of specific loci is commonly performed by PCR enrichment before sequencing [93]. PCR is an effective step for improving the positive rate in the metagenomic sample. Especially when detecting low-abundance foodborne pathogens in foods, the PCR step can significantly improve the accuracy of subsequent nanopore sequencing [94]. However, the targets for microbial enrichment areas are generally limited, which loses the major advantage (untargeted detection) of metagenomics [95]. Additionally, the bias caused by PCR will also have an impact on subsequent sequencing.

Real-time monitoring of nanopore sequencing provides the possibility for efficient enrichment of target sequences. Compared with the original results of second-generation sequencing, which are stored in the form of fluorescent signals that cannot be processed in real-time, nanopore sequencing can monitor sequencing data in real-time through the changing trend of electrical signals. This unique advantage means that reasonable judgments can be made based on the real-time data obtained during the sequencing process. Real-time sequencing monitoring has important applications for specific target sequencing. Adaptive sequencing is a new technology developed based on nanopore sequencing [96]. Adaptive sequencing refers to establishing a decision point during the sequencing process by matching the DNA sequence with the target reference sequence to determine whether the target sequence exists. If the target sequence is present, continue sequencing. Instead, the voltage is reversed, and the DNA strand is ejected to release the nanopore, allowing other strands to enter and continue sequencing. Adaptive sequencing technologies have broad prospects in metagenomics. Instead of screening at the data processing stage, all DNA sequences can be selectively sequenced at the sequencing stage. Therefore, in addition to saving a lot of time for subsequent data analysis, adaptive sequencing can also reduce the impact of non-target genes on real results. Through adaptive sequencing, people can selectively enrich the sequences of foodborne pathogens that may exist in food and the external environment, thereby greatly reducing the sequencing time and rapidly realizing metagenomic analysis. Alexander Payne et al. [97] achieved efficient enrichment of target sequences by nanopore-adaptive sequencing. Moreover, since the principle of adaptive sequencing is completely different from previous enrichment methods, adaptive sequencing can be combined with other enrichment techniques to improve the overall enrichment efficiency. Mingyu Gan et al. [98] combined enzyme-based host depletion and nanopore-adaptive sequencing for the enrichment of microorganisms in clinical samples. Their study showed that the enrichment efficiency of the combination of the two methods was significantly higher than that of the single enrichment method. The diversity of microorganisms detected by the combinatorial methods increased significantly. The above results demonstrated the potential of nanopore-adaptive sequencing to target enriched sequences in clinical samples. However, there are few studies on the detection of foodborne pathogens by adaptive sequencing. This may be because the background genome in food is more complex than clinical samples, and adaptive sequencing requires higher identification accuracy of bases. The detection of foodborne pathogens from food production to food consumption is promising, with the development of adaptive sequencing and the combination with enrichment methods such as depleting the host genome. Furthermore, with the simplicity and speed of nanopore sequencing library preparation and the portability of nanopore instruments, all microorganisms present during food processing or food storage can be temporarily sampled and rapidly sequenced [99]. This avoids possible partial DNA degradation during sample storage and transportation and supports the provision of appropriate recommendations based on sequencing results directly at the sample collection site.

Food commodities are often threatened by multiple foodborne pathogens, not just one. Achieving multiple identifications of viable bacteria is necessary for a good monitoring method [100]. Metagenomic analysis can be applied to monitor all foodborne pathogens present throughout the food supply chain. Ji-Yeon Hyeon et al. [101] used the MinION device to achieve rapid detection and phylogenetic identification of Salmonella in lettuce samples. These authors demonstrated that nanopore sequencing-based metagenomics provides sufficient sequencing depth and sequence read length to allow high-quality genome assembly and detection of key virulence genes. Combining short-term enrichment, immunomagnetic separation, and multiple displacement amplification with nanopore sequencing, Fereidoun Forghani et al. [102] dramatically reduced overall turnaround time. Data generated within 15 h of sequencing supported the detection and serotyping of Salmonella and E. coli in flour. Even if the concentration of target pathogens is low, preliminary analysis of the sample can still be performed within 1 h after enrichment and incubation.

Metagenomics can also be applied to pathogen population-level studies. This will help us understand the association process of pathogenic microorganisms with various environments and reveal the mechanism of interaction between microorganisms. Metagenomics analysis of the microbiota present in the food supply chain can reveal the composition and overall changes of the microbial population. Yang et al. [103] used metagenomic analysis to observe changes in pathogen populations in the beef production chain. They found that the relative abundance of all pathogenic and non-pathogenic bacteria decreased dramatically from the feedlot to the final beef product, but the relative proportions of certain pathogenic species increased in the microbiota remaining in the final product. It is speculated that this may be due to a lack of competition from other bacteria or the ability of related pathogens to survive antimicrobial measures. In fermented foods such as pickles and cheese, metagenomic analysis can provide valuable information to advance understanding of the changing processes of the microbiota within them [104,105]. In addition, metagenomic analysis of gut microbiota has been a research hotspot. Studying the composition and dynamics of the gut microbiota of livestock and poultry will help us understand the role of beneficial bacteria in resisting colonization by foodborne pathogens and how probiotics can improve host health. This may have important applications in improving livestock and poultry production [106].

## 4. Nanopores for the Establishment of Real-Time Monitoring and Traceability System

Establishing an effective surveillance system for foodborne pathogens, rapid and accurate identification of pathogens during foodborne outbreaks, and traceability of food sources are critical to preventing the further spread of infection [107]. In an ideal pathogen surveillance network (Figure 4), generating large numbers of gene sequences needs to be analyzed, investigated, and acted upon quickly [108]. Unfortunately, the existing foodborne pathogen detection network relies on short-read sequencing technology, which has a certain lag and cannot truly analyze and monitor in real-time. Based on the characteristics of nanopore real-time sequencing, it is expected to realize real-time monitoring and tracking of foodborne pathogens and quickly determine the source of pollution and the transmission route of pathogens. Metagenomics plays an important role in surveillance and traceability systems [109]. It has the potential to serve as a new method for foodborne microbial outbreak investigation, source attribution, and risk assessment. Metagenomics analysis can skip the culturing step and quickly identify unknown or emergent microorganisms present in the food or food production chain, completing the screening of all disease-causing pathogens. If the causative pathogen has been speculated based on the clinical symptoms of the patient, it can be combined with adaptive sequencing to sequence the target pathogen to shorten the time of screening and exclusion. Furthermore, nanopore sequencing can monitor antimicrobial resistance genes in foods with complex background genomes. This allows us to obtain information on its prevalence, distribution, and possible routes of transmission in the food production chain. Despite the advantages of nanopore sequencing-based metagenomics in rapid monitoring, the presence of large amounts of genetic material of abiotic origin in food can still interfere with sequencing results. When Noyes et al. [110] attempted to assess AMR determinants in final beef products by metagenomic assessment, they found that the vast majority of reads in meat came from the genomes of slaughtered animals. Furthermore, when metagenomic assays are used to detect virulence or AMR genes, it is difficult to predict whether they belong to a specific pathogen or background microbiota. The single, accurate genetic information provided by WGS can complement metagenomics analysis. Nanopore sequencing-based WGS has the ability to clearly identify SNPs and SVs [111]. All real-time sequence data collected from surveillance systems can be used to perform phylogenetic analysis of various foodborne pathogens in human populations and to infer genetic relationships among subtypes through phylogenetic studies. Genetic relationships can be used for source attribution. On the one hand, genetic relationships may reflect an association with a specific host or resident environment and indicate possible transmission routes for pathogen subtypes. On the other hand, this information is correlated with the frequency of subtypes in clinical isolates to infer the most important food source for a certain subtype of the pathogen to cause disease.

The establishment of a real-time monitoring and traceability system based on nanopore sequencing is conducive to real-time monitoring and early warning of foodborne diseases. Ideally, real-time sequencing data from metagenomic analysis would allow for initial rapid screening and serotyping in the early stages of foodborne pathogens outbreaks. The metagenomics-based database is then complemented by WGS through specific cultures to complete the characterization of the complete characterization of foodborne pathogens and trace the food sources and nodes leading to outbreaks.

## 5. Nanopore Sequencing for Poultry Production Safety

Poultry and related products are an important source of foodborne pathogens such as Campylobacter, Salmonella, Listeria, and others [112]. Chicken is the leading cause of foodborne illness outbreaks in the United States, according to the CDC. Eggs and egg products are also important vectors of foodborne disease outbreaks in Europe. This may be because chickens and other birds are natural hosts for microorganisms such as Salmonella and Campylobacter, which colonize the gut in large numbers. Improper handling during slaughter and subsequent processing can greatly increase the potential for contamination of poultry products with these bacteria [113]. The development of nanopore sequencing technology has opened the possibility for genomic analysis to improve the quality of poultry products. Traditional sequencing methods are difficult to apply to farms on a large scale due to technical and instrument limitations. The portability, low cost, and ease of operation of nanopore sequencing mean that this technology can contribute to improving poultry health, product quality, and genetic disease resistance through WGS or metagenomics analysis. The complex microbiota present in the gastrointestinal tract of animals is closely related to the nutrient metabolism and immune regulation of the host [114]. In addition, the biological barrier constructed by the intestinal flora can prevent the invasion of pathogenic microorganisms from the outside world and the body. Therefore, understanding the mechanisms by which the composition of the gut microbiota interacts with microbes is crucial for the health of animal organisms [115]. Metagenomics plays an important role in the analysis of gut microbes because it is generally believed that the culturable microorganisms in the intestinal flora constitute only 1% of the microbiota. Nanopore sequencing enables rapid macrogenomic analysis of the gut microbial community to reveal the species composition and abundance of the gut microbiota. At the same time, setting data on the gut microbiome of normal poultry as a benchmark for gut health enables the identification of any changes that affect health and disease when the gut flora is dysregulated. Studies have shown that probiotics play an important role in microbial balance [116]. For example, probiotics can promote nutrient absorption, resist or reduce the colonization of harmful bacteria, and inhibit intestinal inflammation to maintain intestinal health. Metagenomics analysis is of great significance in studying the mechanism of action of probiotics in the gut. For example, probiotics and micro-ecological balance. On the one hand, through nanopore sequencing, deficiencies in the gut microbiota of poultry are identified and supplemented by the addition of specific probiotics to the feed to improve gut health. On the other hand, regular metagenomics analysis of the intestinal flora of poultry can enable the administration of specific probiotics to establish an immune barrier and reduce the invasion of harmful bacteria when a large number of foodborne pathogens are colonized [117].

By using nanopore sequencing technology-based metagenomics analysis, the gut microbiota of poultry can be rapidly sequenced at any time. Through sequence analysis, the contribution of the gut microbiota to host health and a new approach to bypass the colonization of the gut by foodborne pathogens to enhance poultry health and ultimately improve poultry product safety and quality may be uncovered.

## 6. Nanopore Sequencing against COVID-19

Coronavirus Disease 2019 (COVID-19) refers to pneumonia caused by severe acute respiratory syndrome coronavirus 2 (SARS-CoV-2). It is transmitted by direct contact and aerosol and can cause clinical symptoms such as fever, dry cough, and fatigue in patients [118]. COVID-19 has spread globally and has caused hundreds of millions of economic losses and millions of deaths. Maintaining social distance and reducing crowd gatherings effectively prevent the large-scale spread of the new Corona epidemic. However, some studies have found that the SARS-CoV-2 virus can survive for more than 3 weeks and remain infectious when the temperature reaches −18 °C [119]. In addition, Peipei Liu et al. [120] isolated SARS-CoV-2 directly from the surface of cod outer packaging for the first time. This means that there is a possibility that SARS-CoV-2 infects humans through cold chain transportation and triggers a new round of epidemics [121]. This potential risk raises concerns about the safety of frozen food and may change the current epidemic prevention policies of various countries against human-to-human transmission and increase the detection of SARS-CoV-2 in imported cold chain food. Nanopore sequencing technology has good application prospects in the fight against COVID-19. In addition, Masateru Taniguchi et al. [122] combined machine learning and nanopores to create an artificial intelligence nanopore. The platform was able to rapidly detect SARS-CoV-2 in saliva samples with 90% sensitivity and 96% specificity through a 5-min measurement. The detection results of this platform may serve as a preliminary judgment standard for the detection of SARS-CoV-2 and provide a reference for subsequent nanopore sequencing.

The ability of nanopore direct RNA sequencing can effectively reduce the time wasted in library preparation and RT-PCR. Additionally, when detecting the possible presence of SARS-CoV-2 on frozen food packaging, nanopore sequencing-based metagenomic analysis can quickly obtain results from the environment and trace the most likely contamination links, preventing further spread of the virus. Most importantly, COVID-19 has now spread globally, but some countries still lack corresponding testing methods. One of the reasons is that second-generation sequencing instruments such as Illumina are generally expensive and require a suitable installation environment. The advantages of nanopore sequencing portability and low cost mean that this emerging technology can play an important role in the detection of local and national SARS-CoV-2. Nanopore sequencing technology provides a new strategy for rapid detection of SARS-CoV-2 in non-laboratory settings. This is helpful for people to quickly check for viruses that may be attached to the surface of refrigerated food and prevent the further spread of SARS-CoV-2 along the cold chain.

## 7. Challenges

The unique advantages of nanopore sequencing have good application prospects in the food industry. However, some existing limitations pose challenges to the large-scale application of nanopore sequencing to the food industry. Firstly, although nanopore sequencing is still improving, the overall error rate of sequencing results is still higher than that of traditional sequencing methods. The error rate of nanopore sequencing correlates with the identification of characteristic electrical signals. On the one hand, the fast DNA translocation speed is the main factor affecting the accuracy of sequencing results [123]. The speed of DNA translocation is too fast to afford the necessary current resolution in a moving strand. The problem of slowing the passage of bases through the nanopore to improve the accuracy of base identification remains to be solved. On the other hand, bioinformatics tools can also affect the accuracy of base calling, genome assembly, and subsequent data analysis [124]. Therefore, the continuous upgrading of bioinformatics tools is one of the effective measures to improve the accuracy of nanopore sequencing. Secondly, nanopore sequencers mainly use biological nanopores to complete sequencing. However, biological nanopores are composed of proteins that may lose activity due to long-term storage or harsh environments, thereby affecting sequencing performance [125]. Solid-state nanopores are expected to be an improved direction for future nanopores due to their stability and reproducibility. However, solid-state nanopores may yield lower signal-to-noise ratios [126]. A low signal-to-noise ratio presents a serious challenge to finding useful signals in the noise. Moreover, the fabrication of solid-state nanopores requires higher precision, which is currently mainly limited by technology.

## 8. Conclusions and Future Perspectives

Foodborne diseases have shown a high incidence in recent years. Detection of possible pathogenic microorganisms in food and processing environments is a better way to reduce the occurrence of foodborne diseases. As a third-generation sequencing technology, nanopore sequencing has the advantages of fast, unlabeled, long read length, real-time sequencing. This article reviews the principles of this technology, the application in WGS and metagenomics, and its contribution to establishing real-time surveillance systems, improving poultry production, and combating COVID-19. It plays an important role in promoting the development of foodborne pathogens detection and studying microbial communities, and triggering a new revolution in food safety.

Although nanopore sequencing technology has shown excellent performance and promise in foodborne pathogens detection, various challenges need to be solved to improve continuously. Much research work is still needed to achieve large-scale applications of nanopore sequencing. At present, the accuracy of nanopore sequencing still needs to be improved. Enhanced identification of characteristic peaks of electrical signals and improved bioinformatics tools may be the future development direction to reduce the error rate of nanopore sequencing. In addition, since solid-state nanopores can be preserved for a long time and reused, they can be an effective tool to replace biological nanopores. However, the technical problems and low signal-to-noise ratio of solid-state nanopores need to be solved. In summary, nanopore sequencing is a technology with great potential, which holds great promise in the detection and typing of foodborne pathogens. The combination of nanopore sequencing and other technologies is beneficial for establishing a pathogen monitoring system based on nanopore sequencing and applying it to the field of food quality control.

## Figures and Tables

**Figure 1 nanomaterials-12-01534-f001:**
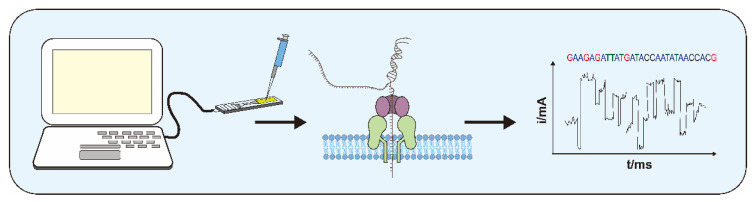
Schematic representation of the principle and process of nanopore sequencing.

**Figure 2 nanomaterials-12-01534-f002:**
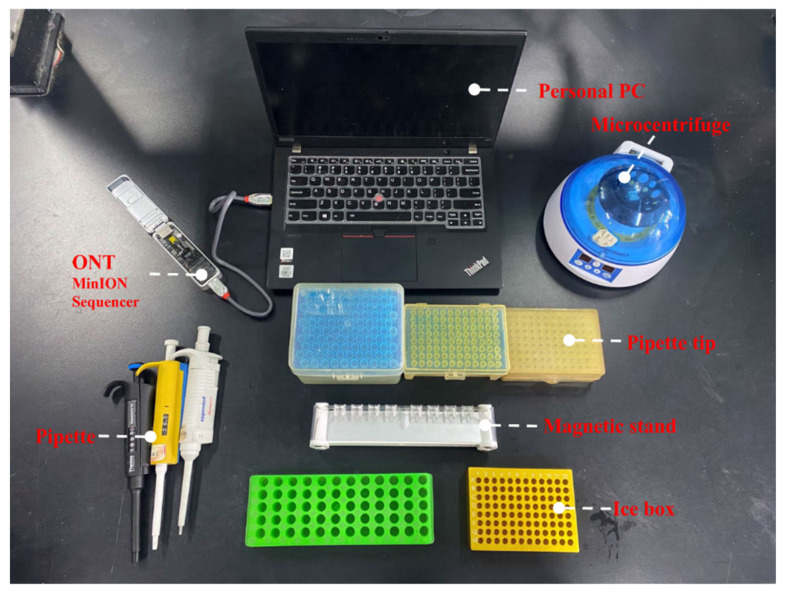
Equipment needed for rapid sequencing with the MinION sequencer.

**Figure 3 nanomaterials-12-01534-f003:**
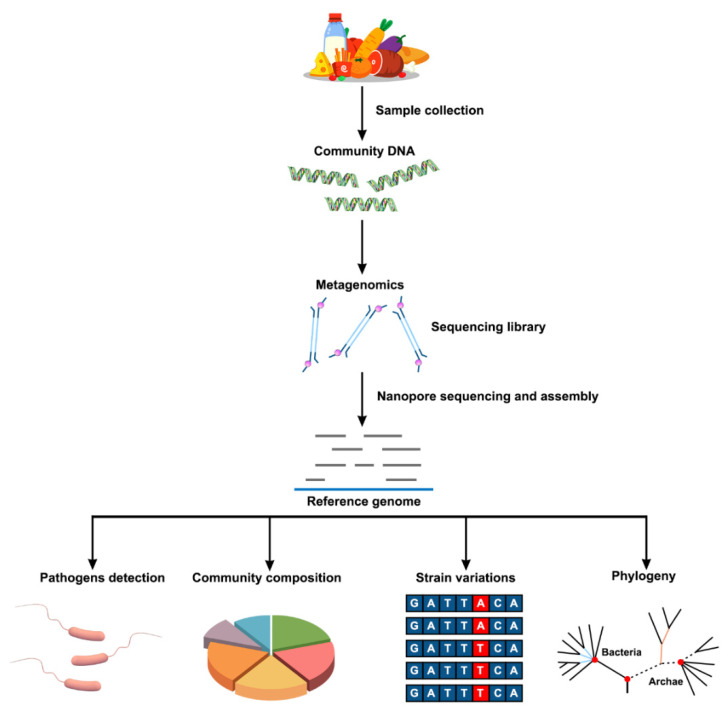
Schematic flow chart of the metagenomic analysis of food samples using nanopore sequencing. After collecting samples from food, total DNA is extracted to obtain community DNA. Subsequently, motor proteins and linkers are added to the resulting DNA to construct a library. In addition, an appropriate library construction kit can be selected to improve the accuracy of sequencing. Then, nanopore sequencing is performed, and the resulting reads are assembled. The resulting gene information can be used for the detection of pathogenic microorganisms, community composition analysis, detection of variation, and phylogenetic analysis.

**Figure 4 nanomaterials-12-01534-f004:**
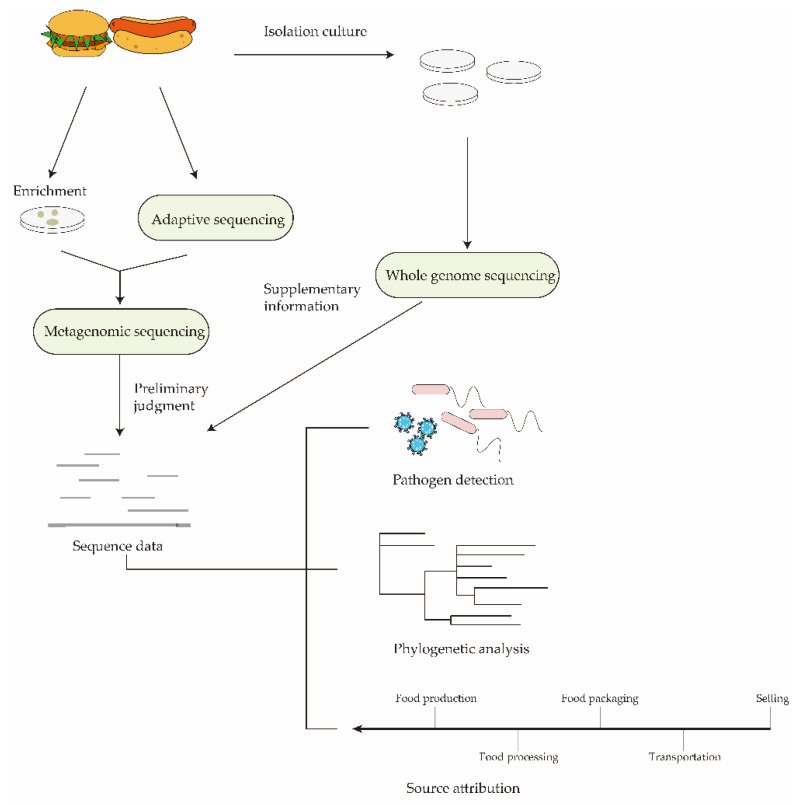
Schematic diagram of an ideal real-time monitoring and traceability system. In the early stages of a foodborne pathogen outbreak, the pathogen can be inferred based on the clinical symptoms of the infected individuals. The associated foodborne pathogens are then selectively enriched from food, or target sequences are enriched directly by adaptive sequencing. Finally, the enriched microorganisms or sequences are subjected to nanopore sequencing. Sequence data generated by sequencing can be used for pathogen detection, phylogenetic analysis, and source attribution. If the amount of data is insufficient, perform WGS after isolation and culture of foodborne pathogens in food. The resulting detailed sequence data can complement the results of metagenomic analyses.

**Table 1 nanomaterials-12-01534-t001:** Main detection methods for foodborne pathogens.

Method	Biomarker	Advantages	Limitations	Ref.
Culture	Bacteria and viruses	Gold standard	Time-consuming,	[19,20]
Polymerase Chain Reaction (PCR)	Nucleic acid	High specificity and low turnaround time.	false-negative and false-positive results.	[21,22,23]
Real-time PCR (qPCR)	Nucleic acid	High specificity and sensitivity, low turnaround time, and quantifiable results.	Expensive equipment, complex operation, and false-positive results.	[24,25,26,27]
Loop-Mediated Isothermal Amplification (LAMP)	Nucleic acid	High specificity and sensitivity, low cost, and easy operation.	False-positive results.	[28,29,30]
Enzyme-Linked ImmunoSorbent Assay (ELISA)	Protein	Fast detection speed, low cost, and easy operation.	False-positive results and short regent life.	[31,32]

**Table 2 nanomaterials-12-01534-t002:** Application of nanopore sequencing in foodborne diseases.

Methods	Applications of Nanopore	Refs.
WGS	Pathogen identification and typing. Detection of AMRgenes. Direct sequencing of RNA and detection of RNA modifications. Adaptive sequencing and targeted detection of pathogens. Microbial community composition and changes. Establishment of real-time monitoring and traceability systems for foodborne pathogens. Pathogen detection, studying the gut microbial population of poultry and the mechanism of action of probiotics. Detection and traceability of SARS-CoV-2 on the surface of the object.	[42]
		[43]
Metagenomics analysis		[44,45]
		[46,47] [48]
Combined WGS with Metagenomics		[49,50] [51,52,53] [54,55]

**Table 3 nanomaterials-12-01534-t003:** Comparison of methods for detection of AMR genes.

Method	Advantages	Limitations	Ref.
Antimicrobial sensitivity testing	Simple operation, intuitive results, and low cost.	Time consuming and difficulty in detecting specific AMR genes.	[68]
PCR	High specificity, low turnaround time, simple operation, and low cost.	False-negative and false-positive results.	[69,70]
DNA microarray technology	Simple operation, low turnaround time, and suitable for detection of large-scale samples.	High cost and low sensitivity.	[71]
WGS	Cover many different targets at the same time and subtype-specific gene variants.	High cost and difficult application for large-scale AMR gene detection.	[72,73,74]

## Data Availability

Not applicable.

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
