# Peer review of "Application of Nanopore Sequencing in the Detection of Foodborne Microorganisms"

_nanomaterials, 2022, doi:10.3390/nano12091534_

Round 1

Reviewer 1 Report

The review is presenting the approach of NGS by the MINIon technology to detect food pathogens. The review is covering the topic very superficially, mentioning numerous times how beneficial is the technique, without covering the actual problematic issues, and potential weaknesses. For example:

  1. the issue of sample preparation has to be covered, emphasizing the need to deplete the host genome to lower the background
  2. the issue of initial amplification of the pathogens prior to the NGS should be covered
  3. the issue of sensitivity by NGS detection directly, or after amplification
  4. the issue of sensitivity and limit of detection in raw food as compared to other diagnostic assays.

Author Response

Response to Reviewer 1 Comments

The review is presenting the approach of NGS by the MINIon technology to detect food pathogens. The review is covering the topic very superficially, mentioning numerous times how beneficial is the technique, without covering the actual problematic issues, and potential weaknesses. For example:

Point 1: the issue of sample preparation has to be covered, emphasizing the need to deplete the host genome to lower the background

Response 1: We are very grateful for the comment and suggestion from reviewer #1. We added sentences describing that depletion of the host genome is still necessary for nanopore sequencing in the revised manuscript (references No.69-71, 78, 79, line 291-296, line 328-344, pages 8-9).

Point 2: the issue of initial amplification of the pathogens prior to the NGS should be covered

Response 2: We greatly appreciate your comments and suggestions. We supplemented that the samples prior to sequencing need to undergo enrichment or PCR-targeted amplification. (references No.73-76, line 298-308, page 8)

Point 3: the issue of sensitivity by NGS detection directly, or after amplification

Response 3: We appreciate your valuable comments and suggestions. We supplemented changes in nanopore sequencing sensitivity after PCR amplification. Results showed improved sensitivity and specificity for nanopore sequencing after targeted enrichment. (references No.72, 75, line 296-298, line 302-305, page 8)

Point 4: the issue of sensitivity and limit of detection in raw food as compared to other diagnostic assays.

Response 4: We are very grateful for the comment and suggestion from you. We added the difficulty of the accuracy of nanopore sequencing compared to qPCR and culture methods and explained possible reasons for the decline in nanopore sequencing accuracy. (line 187-197, page 5).

Reviewer 2 Report

I have enjoyed its reading and gladly recommend it to my collegues. Meanwhile, minor corrections in English punctuation are required:

wrong sentence breaks (e.g., missing predicate etc.) as

lines 74-76: Detection of pathogenic microorganisms that may exist in food processing links and food is an effective means to prevent foodborne diseases[30]. On the other hand, investigate foodborne illnesses that have emerged and link them to contaminated food.

lines 165-168 :  Long reads provide a more definitive way to align and match DNA or RNA sequences, providing high-quality, more complete, and contiguous genome assemblies[44]. Not only saves the time required for splicing, but also provides higher quality data for  subsequent genome analysis. There ....

lines 208-210:  Therefore, Hellyer et al. [53] propose that using RNA instead of DNA as a biomarker of bacterial survival may be a better option. Because RNA has a shorter half-life than DNA in an environment of cell inactivation. Nanopore ...

lines  423-425: This means that there is a possibility that SARs-Cov-2 infects humans through cold chain transportation and triggers a new round of epidemics[86]. Raising concerns about the safety of frozen food. This ...

Unclear phrase inline 102: ....dropped into the instrument, the charged properties of the molecules themselves will .... (charged molecules?)

Mispirint in line 481: Conflicts of Interest: The authors declare that they are no conflict of interest. (there is ..?)

Author Response

Response to Reviewer 2 Comments

wrong sentence breaks (e.g., missing predicate etc.) as

Point 1: lines 74-76: On the other hand, investigate foodborne illnesses that have emerged and link them to contaminated food.

Response 1: We are very grateful for the comment and suggestion from reviewer #2. We modified and refined the sentence, ‘’On the other hand, emerging foodborne illness findings can be linked to contaminated food. This connection…….’’ (line 87-89, page 3).

Point 2: lines 165-168 : Not only saves the time required for splicing, but also provides higher quality data for subsequent genome analysis.

Response 2: We appreciate your valuable comments and suggestions. We have revised the English language questions, ‘’It not only saves time required for splicing but also provides higher quality data for subsequent genomic analysis.’’ (line 183-184, page 5)

Point 3: lines 208-210: Because RNA has a shorter half-life than DNA in an environment of cell inactivation

Response 3: We are very grateful for the comment and suggestion from you. We revised the sentence, “Hellyer et al. [59]. propose that using RNA instead of DNA as a biomarker of bacterial survival may be a better option because RNA has a shorter half-life than DNA in an environment of cell inactivation.’’ (line 235-236, page 7)

Point 4: lines  423-425: Raising concerns about the safety of frozen food.

Response 4: We are very grateful for your valuable comments and suggestions. We modified and refined the sentence, ‘’This potential risk raises concerns about the safety of frozen food.’’(line 497-498, page 13)

Point 5: Unclear phrase inline 102: ....dropped into the instrument, the charged properties of the molecules themselves will .... (charged molecules?)

Response 5: We are very grateful for the comment and suggestion from you. We modified and refined the sentence, “DNA and other molecules will move towards the nanopore under the effect of an electric field after they are dropped into the sequencer.’’ (line 114-116, page 3)

Point 6: Mispirint in line 481: Conflicts of Interest: The authors declare that they are no conflict of interest. (there is ..?)

Response 6: We appreciate your valuable comments and suggestions. We revised the “they’’ to ‘’there’’(line 557, page 14)

Reviewer 3 Report

The authors review nanopore sequencings for the detection of foodborne microorganisms. As discussed in the manuscript, nanopore platform is a powerful tool for analyzing DNA/RNA and the technology has been expected to be used practically in some field such as medical, healthcare, and public health. In that sense, this review is a valuable paper and will help people who start to work on nanopore devices and foodborne microorganisms. Thus, I would like to recommend this review paper as an acceptable manuscript for nanomaterials. However, authors should revise following points before accepting.

  1. Although, as authors claim, DNAs have one of the best information for analyzing microorganisms from the viewpoint of high accuracy, nanopore devices recently enable us to detect microorganisms such as E. coli directly by analyzing shape of them [1], and the accuracy has been improved by using Machine learning [2]. In addition, these shape analysis nanopore devices are applied for a nanobiosensor [3]. Therefore, the authors should explain these nanopore platforms in the introduction because this paper is a review paper.

[1] Nanotechnology 28, 155501 (2017).

[2] Scientific Reports volume 10, Article number: 15525 (2020)

[3] Anal. Chem. 93, 7037 (2021).

  1. I don't think there are big points which should be revised with the main text. However, in 6. Nanopore sequencing against COVID-19, the authors should refer to the following paper [4].

[4] “Combining machine learning and nanopore construction creates an artificial intelligence nanopore for coronavirus detection”,

https://www.nature.com/articles/s41467-021-24001-2

  1. There are too few figures, so it would be better to add other figures. The authors should explain more visually.

Author Response

Response to Reviewer 3 Comments

The authors review nanopore sequencings for the detection of foodborne microorganisms. As discussed in the manuscript, nanopore platform is a powerful tool for analyzing DNA/RNA and the technology has been expected to be used practically in some field such as medical, healthcare, and public health. In that sense, this review is a valuable paper and will help people who start to work on nanopore devices and foodborne microorganisms. Thus, I would like to recommend this review paper as an acceptable manuscript for nanomaterials. However, authors should revise following points before accepting.

Point 1: Although, as authors claim, DNAs have one of the best information for analyzing microorganisms from the viewpoint of high accuracy, nanopore devices recently enable us to detect microorganisms such as E. coli directly by analyzing shape of them [1], and the accuracy has been improved by using Machine learning [2]. In addition, these shape analysis nanopore devices are applied for a nanobiosensor [3]. Therefore, the authors should explain these nanopore platforms in the introduction because this paper is a review paper.

Response 1: We are very grateful for the comment and suggestion from reviewer #3. We have added descriptions and functions of these nanopore platforms and cited relevant articles. (line 50-57, page 2)

Point 2: In 6. Nanopore sequencing against COVID-19, the authors should refer to the following paper [4].

Response 2: We appreciate your valuable comments and suggestions. We complemented the role played by the nanopore platform in the detection of SARS-CoV-2 and provided relevant articles.(line 501-506, page 13)
